# Test–Retest Reliability, Agreement and Criterion Validity of Three Questionnaires for the Assessment of Physical Activity and Sedentary Time in Patients with Myocardial Infarction

**DOI:** 10.3390/ijerph20166579

**Published:** 2023-08-15

**Authors:** Marcus Bargholtz, Madeleine Brosved, Katarina Heimburg, Marie Hellmark, Margret Leosdottir, Maria Hagströmer, Maria Bäck

**Affiliations:** 1Department of Medicine, Lindesberg Hospital, 711 82 Lindesberg, Sweden; 2Department of Occupational Therapy and Physiotherapy, Sahlgrenska University Hospital, 413 45 Gothenburg, Swedenmaria.back@liu.se (M.B.); 3Department of Clinical Sciences Lund, Neurology, Skane University Hospital, Lund University, 222 42 Lund, Sweden; 4Department of Physiotherapy, Orebro University Hospital, 701 85 Orebro, Sweden; 5Department of Cardiology, Skane University Hospital, 214 28 Malmo, Sweden; 6Department of Clinical Sciences Malmo, Lund University, 214 28 Malmo, Sweden; 7Department of Neurobiology, Care Sciences and Society, Division of Physiotherapy, Karolinska Institutet, 171 77 Stockholm, Sweden; 8Academic Primary Health Care Centre, Region Stockholm, 113 65 Stockholm, Sweden; 9Department of Medical and Health Sciences, Division of Physiotherapy, Linkoping University, 581 83 Linkoping, Sweden

**Keywords:** accelerometer, cardiac rehabilitation, myocardial infarction, physical activity, sedentary time, self-report

## Abstract

Regular physical activity (PA) and limited sedentary time (SED) are highly recommended in international guidelines for patients after a myocardial infarction (MI). Data on PA and SED are often self-reported in clinical practice and, hence, reliable and valid questionnaires are crucial. This study aimed to assess the test–retest reliability, criterion validity and agreement of two PA and one SED questionnaire commonly used in clinical practice, developed by the Swedish National Board of Health and Welfare (BHW) and the Swedish national quality register SWEDEHEART. Data from 57 patients (mean age 66 ± 9.2 years, 42 males) was included in this multi-centre study. The patients answered three questionnaires on PA and SED at seven-day intervals and wore an accelerometer for seven days. Test–retest reliability, criterion validity and agreement were assessed using Spearman’s rho and linearly weighted kappa. Test–retest reliability was moderate for three of the six-sub questions (k = 0.43–0.54) within the PA questionnaires. For criterion validity, the correlation was fair within three of the six sub-questions (r = 0.41–0.50) within the PA questionnaires. The SED questionnaire had low agreement (k = 0.12) and criterion validity (r = 0.30). The studied questionnaires for PA could be used in clinical practice as a screening tool and/or to evaluate the level of PA in patients with an MI. Future research is recommended to develop and/or evaluate SED questionnaires in patients with an MI.

## 1. Introduction

Cardiovascular disease (CVD) is the most common cause of death worldwide [1]. Regular physical activity (PA) and limited sedentary time (SED) are strongly recommended, in terms of both the primary [2] and secondary prevention of CVD [3]. Individuals with CVD are recommended to engage in moderate-intensity physical activity (MPA), 150–300 min/week, or vigorous-intensity physical activity (VPA), 75–150 min/week, or a combination of both, and to limit SED [3]. Previous PA recommendations included bouts of 10 min or more of PA, which have now been omitted due to new evidence that supports the belief that any bout length of PA contributes to health benefits [4]. There is an associated risk of low levels of PA, high levels of SED and an increased all-cause and CVD mortality [5,6,7,8,9].

One way of screening the levels of PA and SED is through self-reported questionnaires, which are commonly used in clinical settings because they are easy to use and cost efficient [10]. Although questionnaires are widely implemented, they are prone to reporting bias. For example, individuals can have difficulty remembering their level of PA and SED over time [11]. Objective instruments for measuring PA and SED include the accelerometer. Accelerometers detect acceleration in up to three directions and are able to determine the quantity and intensity of movement [12]. They have high validity when measuring PA [13], but they are time consuming and expensive [10]. It is crucial that self-reported questionnaires are valid and reliable when assessing levels of PA and SED. Strong test–retest reliability is important to show that a questionnaire is free from measurement error and is stable over time [14].

The Swedish Web-system for Enhancement and Development of Evidence-based care in Heart disease Evaluated According to Recommended Therapies (SWEDEHEART) quality register continuously provides information on patient care needs, treatments and treatment outcomes after an acute myocardial infarction (MI) [15]. SWEDEHEART includes a questionnaire consisting of two questions on PA (Haskell-Q), which are based on the highly cited PA recommendations to promote and maintain health by Haskell et al. [16]. This questionnaire was included in the SWEDEHEART register in 2016 and is administered by physiotherapists nationwide before and after the exercise-based cardiac rehabilitation programme. The Swedish National Board of Health and Welfare (BHW) recommends different questionnaires to assess levels of PA in health care (BHW-Q) [17] and time spent as SED (GIH-SED-Q) [18,19]. The BHW-Q is intended to identify people that are insufficiently physically active and to examine the level of PA after intervention.

In a recent single-centre study, the convergent validity of the BHW-Q, Haskell-Q and GIH-SED-Q was assessed in patients with an MI [20]. Although the Haskell-Q is broadly used in clinical practice among patients with an MI and the BHW-Q and GIH-SED-Q are recommended for use as screening tools in health care, the test–retest reliability, agreement and criterion validity of these questionnaires among patients with an MI are still not known. The purpose of the present study was therefore to assess the test–retest reliability, criterion validity and agreement of the BHW-Q, Haskell-Q and GIH-SED-Q in patients after an MI.

## 2. Materials and Methods

### 2.1. Study Design and Participants

This multi-centre study was performed at five hospitals in Sweden: Lindesberg Hospital (LBG), Sahlgrenska University Hospital (SU), Skane University Hospital Malmo/Lund (SU-M/SU-L) and Orebro University Hospital (USO) between April and December 2021. The inclusion criteria were a diagnosis of MI, age < 80 years and registered in the SWEDEHEART register. The exclusion criteria were severe physical or mental illness preventing the performance of the tests and difficulty understanding spoken and written Swedish. This study was approved by the Regional Ethical Review Board at the University of Uppsala (registration number 2020-05142 with amendment 2021-01634).

### 2.2. Procedure

Patients were recruited consecutively from coronary care units and cardiac rehabilitation centres. On the first visit to the physiotherapist, about two to four weeks after the MI, patients were informed and asked about participation in the study and provided their written informed consent prior to being enrolled in the study. After being enrolled, the patients were asked to complete the BHW-Q, Haskell-Q, GIH-SED-Q and baseline characteristics, including age, gender, co-morbidity, civil status, education and occupation status. The patients were given a new set of questionnaires together with an ActiGraph GT3X accelerometer (ActiGraph, Pensacola, FL, USA) and a wear-time diary to take home. The patients were asked to wear the accelerometer on their hip during waking hours for seven consecutive days. They were instructed to take the accelerometer off during water-based activities (e.g., showering or swimming) and to self-register their wear time in a diary. After seven days, the patients completed the questionnaires once again and send them back, in a pre-paid envelope, together with the accelerometer and diary, to their respective CR centre. The choice of a test–retest interval of seven days is considered appropriate when evaluating questionnaires with a recall time of the previous and/or a usual week [14].

### 2.3. Physical Activity and Sedentary Time Questionnaire

A total of two questionnaires on PA (BHW-Q, Haskell-Q) and one on SED (GIH-SED-Q) were included in this study. Two supplementary adapted PA questions were added to this study, relating to the Haskell-Q (MPAtot) and BHW-Q (MPAtot) without 10 min bouts. This was due to the recently updated guidelines for PA where recommendation for bouts of 10 min or more has been omitted [2,4].

#### 2.3.1. Haskell-Q

The Haskell-Q focuses on the frequency of PA during the previous week [16]. Moderate and vigorous intensity are separated. The patients are asked to mark the number of days (0–7) that correspond to the indicated activity level for each question.

Haskell-Q MPA: “Number of days (0–7) with moderate-intensity physical activity accumulated towards the 30-min minimum by performing bouts each lasting 10 or more minutes during the last week”.Haskell-Q VPA: “Number of days (0–7) performing vigorous-intensity physical activity/exercise for a minimum of 20 min”.Haskell-Q MPAtot (supplementary question): “Number of days (0–7) with moderate-intensity physical activity accumulated toward the 30-min minimum during the last week”.

#### 2.3.2. BHW-Q

The BHW-Q focuses on time spent in VPA and MPA defined as categorical answer options during a usual week (Figure 1). The BHW-Q generates a continuous outcome of total PA, called “activity minutes”. “Activity minutes” are the number of minutes of VPA × 2 + the number of minutes of MPA [17]. 

BHW-Q VPA: “During a regular week, how much time do you spend exercising on a level that makes you short winded, for example running, fitness class, or ball games?”.BHW-Q MPA: “During a regular week, how much time are you physically active in ways that are not exercise, for example walks, bicycling, or gardening? Add together all activities lasting at least 10 min”.BHW-Q MPAtot (supplementary question): “During a regular week, how much time are you physically active in ways that are not exercise, for example walks, bicycling, or gardening? Add together all the time”.

#### 2.3.3. GIH-SED-Q

The GIH-SED-Q focuses on time spent while sedentary during a usual day according to categorical answer options (Figure 1).

GIH-SED-Q sedentary time: “How much time do you sit during a usual day, excluding sleep?”.

### 2.4. Criterion Validity Instrument

The three-axial ActiGraph GT3X accelerometer (ActiGraph, Pensacola, FL, USA) was used objectively to measure PA and SED. The ActiLife software, version 6.13.4, was used to initialise, extract and analyse the raw data from the accelerometer. A sampling rate of 30 Hz was used and the raw data were converted to 60 sec epochs in the counts per minute (cpm) unit for the vector magnitude (VM) that combines cpm from three axes into one outcome defined as √ (x^2^ + y^2^ + z^2^). Non-wear time was calculated using the Choi algorithm [21]. Non-wear time was verified by time recorded in the wear-time diaries. At least 540 min and at least one valid day were required to be included in the analysis. The Sasaki [22] cut-points for the VM were used: sedentary is defined as >149 cpm; low PA (LPA) as 150–2689 cpm; moderate intensity PA (MPA) as 2690–6166 cpm and vigorous intensity PA (VPA) as ≥6167 cpm. The weekly mean and median period in wear time, LPA, MPA, VPA and SED, was calculated by dividing the sum of each variable by valid days and multiplying it by 7. “Activity minutes” were calculated by multiplying accelerometer data, VPA, by two, with the addition of MPA. Furthermore, the continuous accelerometer data were categorised into the same categories as the BHW-Q and GIH-SED-Q answer options. Accelerometer data were presented, at group level, as both the median (interquartile range (IQR)) and the mean ± one standard deviation (SD).

### 2.5. Statistical Analyses

Statistical analyses were performed using SPSS, version 28.0 (IBM Corp. Armonk, NY, USA). All the variables were tested for normal distribution using the Shapiro–Wilk test. Nominal data were presented in absolute numbers and percentages. To analyse the test–retest reliability and correlation, a linearly weighted kappa coefficient (k) and Spearman’s rho (r) were used. The k varies from 0 to 1, where 1 is equivalent to perfect agreement and 0 is no agreement. In this study, an agreement of >0.8 was interpreted as excellent, 0.61–0.8 as substantial, 0.41–0.60 as moderate, 0.21–0.40 as fair and 0.01–0.2 as slight agreement [23]. Spearman´s rho correlation coefficient is able to adopt values between −1 and 1, where 1 indicates a perfect correlation and −1 indicates a perfect negative correlation. The closer the coefficient is to 0, the weaker is the correlation. In this study, a correlation was interpreted as very strong (r > ±0.8), moderately strong (r ± 0.6–0.8), fair (r ± 0.3–0.6) and poor (r < ±0.3) [24]. To analyse criterion validity and agreement between objective and subjective measurements, Spearman´s rho (r) and a linearly weighted kappa coefficient (k) were used. For comparisons at group level, an accepted weighted kappa and Spearman’s rho should exceed 0.7 and 0.5, respectively [14].

## 3. Results

### 3.1. Study Population

In all, 238 patients were assessed consecutively for eligibility. A total of 81 patients were included in the study. Of these, five patients were subsequently excluded because they met the criteria for exclusion (n = 4, not being included in SEPHIA; n = 1, having a physical illness preventing them from performing the tests). In addition, 19 patients were excluded due to not having a complete set of either questionnaire data and/or accelerometer data. In all, 57 patients were included in the analysis. The number of patients included from the different hospitals were LBG: n = 10; SU: n = 17; SU-L: n = 8; SU-M: n = 4; and USÖ: n = 18.

Baseline characteristics are shown in Table 1. The majority of the patients were male, had a median age of 69 years and were overweight (mean BMI ≥ 27.6 kg/m²).

The accelerometer data revealed that 74% of the patients reached the current recommendations on PA, i.e., spent at least ≥ 150 min of MVPA, all minutes included. However, with the 10 min bout restriction, only 49% of the patients achieved at least ≥ 150 min of MVPA. A total of 54% of patients spent more than eight hours/day in SED. No significant differences are shown in patients´ self-reported MPA, VPA and SED between the two measurement occasions (Table 2). The patients spent most of their waking time in SED (mean: 479 ± 88 min/day) and second most in LPA (Table 3). 

### 3.2. Test–Retest Reliability

Test–retest reliability was moderate for Haskell-Q MPA (k = 0.434), BHW-Q VPA (k = 0.521), “Activity minutes” without ≥ 10 min bouts (k = 0.442) and GIH-SED-Q (k = 0.535). The remaining questions presented a fair test–retest reliability (k = 0.219–0.375) (Table 4).

### 3.3. Criterion Validity and Agreement

Correlation between PA and SED questions and categorical accelerometer data was fair for Haskell-Q MPA (r = 0.407), BHW-Q MPA (r = 0.578), BHW-Q MPAtot (r = 0.501) and “Activity minutes” (r = 0.429). A fair agreement was shown by BHW-Q MPA and BHW-Q MPAtot (k = 0.341 resp. 0.237). All other questions presented a slight to poor correlation to categorical accelerometer data (r = 0.139–0.291) (Table 5).

## 4. Discussion

To our knowledge, this is the first study to assess test–retest reliability for the Haskell-Q and BHW-Q questionnaires. The main findings were that PA and SED questions in patients with an MI displayed fair to moderate test–retest reliability, with poor to fair correlation and slight to fair agreement assessed with an accelerometer.

The test–retest reliability for the Haskell-Q questionnaire was fair to moderate, with weighted kappa values between 0.32 and 0.43, which is below the recommended level of k = 0.7 [14]. The highest reliability was found for MPA, with at least 10 min bouts, but, overall, there was no large difference between MPA, MPAtot and VPA. Regarding test–retest reliability within the BHW-Q, the highest value was found for VPA (k = 0.63). One reason why test–retest values, in general, are under the recommended k = 0.7 [14] for both the Haskell-Q and BHW-Q questionnaires could be the design of the questionnaires. In the Haskell-Q questionnaire, patients are asked to rate the frequency of physical activity during the latest week and this could vary for natural reasons, for example, temporary illness. In the BHW-Q, on the other hand, patients are asked about physical activity and exercise in a regular week. In theory, it can be hypothesised that the BHW-Q would have a higher test–retest reliability than the Haskell-Q, due to its design. This indicates that the patients in the present study may refer to recall period other than a “usual week” when self-reporting their PA with the BHW-Q. The higher agreement for the BHW-Q VPA is in line with results from previous studies showing that patients generally have the same exercise routine between weeks [25]. On the other hand, activities on a moderate-intensity level may differ more between test occasions and, in addition, patients may not consider that the MPA questions are asking about a regular week. The highest correlation between the Haskell-Q questionnaire and the accelerometer is shown for MPA, with at least 10 min bouts (r = 0.41), which is slightly higher compared with another study in the same context (r = 0.28) [20]. On the other hand, Lönn et al. [20] had a higher correlation between the Haskell-Q VPA and accelerometer data (r = 0.24) than the present study (r = 0.14). The correlation between the Haskell-Q MPA and the accelerometer was higher compared with the Haskell-Q MPAtot, which may suggest that it is challenging for patients to self-report the moderate-intensity level when all the time is included. Another explanation may be that the accelerometer is sensitive to short interruptions in bouts of activity that might not be perceived as an interruption by the patients.

The correlation for the BHW-Q questionnaire and accelerometer was poor to fair in this study. There was only a slight difference between the BHW-Q MPA and BHW MPAtot, and both values are in line with the recommended correlation of r = 0.5 [14]. This suggests that both questions are sufficiently valid to measure moderate-intensity PA. The associations between the BHW-Q MPA and the accelerometer were, however, higher in the present study as compared to the study by Lönn et al. [20]. Moreover, in terms of “activity minutes”, the association between the BHW-Q MVPA/MVPAtot and the accelerometer was higher in our study compared with a study by Olsson et al. [17] in the general population.

Generally, the test–retest reliability in the present study is in line with an earlier larger review of test–retest reliability for different PA questionnaires and populations (k = 0.32–0.87) [26]. Only a few studies have examined the criterion validity of self-reported PA and accelerometer for patients with CVD [20,27,28] and the agreement between PA questions and accelerometer is generally low [28,29].

The test–retest reliability of the GIH-SED-Q was moderate in the present study, which is lower than in the study by Larsson et al. [19], who studied a healthy elderly population (≥65 years) living in a large city region. The correlation between the GIH-SED-Q and the accelerometer is similar to Lönn et al.’s [20] results, r = 0.37 (vs. 0.29). Both correlations are lower compared with the study by Kallings et al. [18] of the GIH-SED-Q and stationary time in a general population, r = 0.48 [18]. On the other hand, the correlations were higher in our study compared with other studies that have examined the validity of SED questions in a CR context (r = 0.14–0.21) [27,28,30]. The agreement was slight for the GIH-SED-Q assessed using the accelerometer in our study, which is in line with the results of Lönn et al. (0.09) [20]. The GIH-SED-Q is limited by the specific formulation about sitting time, which is regarded as sedentary time. Gardner et al. [31] have suggested that people rarely reflect on sitting as a behaviour but more as a by-product of more meaningful sitting activities, which means that self-reported sedentary time may be misclassified [31]. In a systematic review and meta-analysis by Bakker et al. [32] assessing SED in healthy adults, the correlation between the accelerometer and self-reports was strongest for logs and diaries compared with questionnaires (0.63 vs. 0.35). However, there was no major difference between multi-item questionnaires and few-item questionnaires (0.37 vs. 0.34) [32], which suggests that one-item questionnaires may be preferable in a large population, due to time consumption when answering larger multi-item questionnaires. There is, however, a need for future research to further elaborate on the design of sufficiently valid SED questionnaires.

In line with the updated PA recommendations, we assessed the reliability and validity of self-reported PA with and without at least 10 min bouts and found no meaningful difference in outcome within the BHW-Q. Another study by Kambic et al. [28] also found no meaningful difference in self-reported PA with and without at least 10 min bouts using The International Physical Activity Questionnaire short form (IPAQ-SF) when examining validity and agreement compared with an accelerometer [28]. However, regarding the Haskell-Q, the original Haskell-Q MPA with at least 10 min bouts was superior to the adapted Haskell-Q total time, even though the differences were small.

### Methodological Considerations

One strength of this study is the external validity as data were collected from several CR centres in Sweden. The proportion of women included in this study reflects the general MI population in Sweden [33]. A strength of the present study was the inclusion of participants representing a wide range of physical activity levels, i.e., low to high [34,35]. A priori sample size calculation was not performed. A retrospective sample size calculation based on a desired reliability coefficient of 0.7 with a CI of 0.1 has been conducted. With a 50% assurance probability and two testing sessions, a minimum sample size of 101 participants was required. Additional studies with a larger sample size to reach sufficient power are recommended to replicate our findings.

The choice of an appropriate criterion measurement in validation studies has previously been discussed. The doubly labelled water method (DLW) is the gold standard for estimating energy expenditure in free-living individuals [36]. However, the DLW is unable to provide data on the intensity, frequency and pattern of PA or SED. As a result, accelerometer measurements have been suggested as one of the best criterion instruments and the accelerometer is the most used objective instrument when validating self-report PA instruments [37,38,39]. The accelerometer measurements were performed for the same period as the Haskell-Q. We, therefore, have no reason to believe that the patients did not refer to the same period. This is in contrast to the BHW-Q and GIH-SED-Q that consider a usual week.

Measuring PA with accelerometers does have some limitations. For example, it cannot be used in water-based activities (swimming) and activities such as strength training. Bicycling may be reported as a false low activity as the accelerometer is placed on the hip. As accelerometer measurements were performed during an exercise-based CR period where ergometer bicycle exercise is a common exercise modality within exercise-based cardiac rehabilitation, this could be one explanation of the weaker correlations between the accelerometer and self-reported VPA. Accelerometers gather information about acceleration in different directions and do not consider body position. Sedentary behaviour was assessed using a hip-placed accelerometer in terms of energy expenditure and not sedentary condition defined by both energy expenditure and posture. A hip-placed accelerometer has a tendency to misclassify different body positions, such as lying, sitting and standing [40]. This may lead to a discrepancy between SED time of accelerometer data when compared with self-reported SED time. Nonetheless, the accelerometer is frequently used in studies that measure SED [32]. Cut-off values for SED and different PA intensities used in this study are in line with a previous study of the CVD population [28].

## 5. Conclusions

The main findings in this study were that the BHW-Q has a higher criterion validity than the Haskell-Q and that the test–retest reliability is similar for both questionnaires. These PA questionnaires could be used in clinical practice as a screening tool and/or to evaluate the level of PA patients with an MI. On the other hand, as the criterion validity between the GIH-SED-Q and the accelerometer was poor, the further development of the design of sufficiently valid SED questionnaires in patients with an MI is recommended. 

## Figures and Tables

**Figure 1 ijerph-20-06579-f001:**
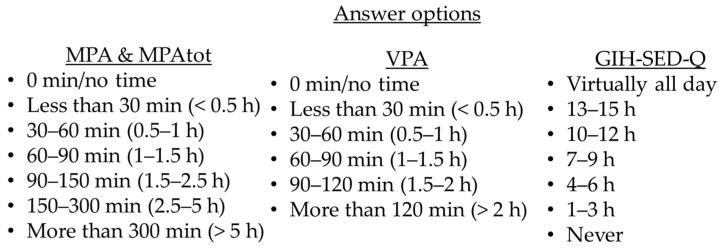
BHW-Q and GIH-SED-Q answer options.

**Table 1 ijerph-20-06579-t001:** Baseline characteristics of study population (n = 57).

Gender, female, n (%)	15 (26)
Age, years, mean (SD)	66 (9.2)
Age, years, median (IQR)	69 (60, 72)
Height, cm (SD)	175 (10)
Weight, kg (SD)	84 (15.4)
BMI, kg/m^2^ (SD)	27.6 (5.3)
Current smoking	5 (9)
Previous smoker > 1 month	23 (41)
DM	4 (7)
Hypertension	23 (40)
Previous MI	8 (14)
Heart failure	5 (9)
Angina pectoris	4 (7)
Atrial fibrillation	4 (7)
Disease in the musculoskeletal system	6 (11)
Other	13 (23)

SD: standard deviation; IQR: interquartile range; BMI: body mass index; DM: diabetes mellitus; MI: myocardial infraction.

**Table 2 ijerph-20-06579-t002:** Descriptive data from questionnaires at test 1 and test 2.

Variable	Test 1	Test 2
Haskell-Q MPA (days/w)	5 (4–7)	5 (3–7)
Haskell-Q VPA (days/w)	1 (0–3)	2 (1–5)
Haskell-Q MPAtot * (days/w)	5 (3–7)	5 (2–6)
BHW-Q VPA (min/w)	45 (0–105)	45 (0–105)
BHW-Q MPA (min/w)	225 (75–300)	225 (45–300)
BHW-Q MPAtot * (min/w)	225 (120–300)	225 (120–300)
GIH-SED-Q (hours/day)	5 (5–8)	5 (5–8)
BHW activity minutes **	300 (143–435)	285 (139–386)
BHW activity minutes ***	300 (188–435)	300 (225–450)

Median (IQR (interquartile range)); MPA: moderate-intensity physical activity; VPA: vigorous-intensity physical activity; SED: sedentary time. * Without ≥10 min bouts; ** BHW-Q VPA × 2 + BHW-Q MPA; *** BHW-Q VPA × 2 + BHW-Q MPA*.

**Table 3 ijerph-20-06579-t003:** Descriptive data from accelerometer.

Variable	Mean (SD)	Median (IQR)
Total wear time (min/day)	845 (66)	847 (794–878)
VM CPM	558 (201)	512 (408–681)
LPA (min/w)	2238 (583)	2303 (1873–2605)
MPA (min/w)	301 (191)	256 (135–402)
VPA (min/w)	21 (56)	1 (0–10)
MVPA ≥ 10 min bouts (min/w)	162 (178)	115 (18–237)
Accelerometer activity minutes ^1^	342 (248)	299 (143–464)
Accelerometer activity minutes ^2^	204 (253)	143 (25–288)
SED (min/day)	479 (88)	486 (43–543)

SD: standard deviation; IQR: interquartile range; VM CPM: vector magnitude counts per minute; LPA: light-intensity physical activity; MPA: moderate-intensity physical activity; VPA: vigorous-intensity physical activity; MVPA: moderate- to vigorous-intensity physical activity; SED: sedentary time; ^1^ VPA × 2 + MPA (min/w). ^2^ VPA × 2 + MVPA ≥ 10 min bouts (min/w).

**Table 4 ijerph-20-06579-t004:** Test–retest reliability and correlation of the questionnaires.

Question	*n*	Kappa (95% CI)	*p* (k)	Spearman’s rho	*p* (rho)
Haskell-Q MPA	56	0.434 (0.284–0.585)	<0.001	0.573 *	<0.001
Haskell-Q VPA	56	0.317 (0.152–0.482)	<0.001	0.409 *	0.002
Haskell-Q MPAtot ^1^	55	0.375 (0.220–0.530)	<0.001	0.532 *	<0.001
BHW-Q VPA	56	0.521 (0.359–0.684)	<0.001	0.633 *	<0.001
BHW-Q MPA	56	0.295 (0.104–0.485)	0.007	0.345 *	0.009
BHW-Q MPAtot ^1^	55	0.337 (0.139–0.535)	<0.001	0.383 *	0.004
BHW activity minutes ^2^	55	0.354 (0.197–0.512)	<0.001	0.542 *	<0.001
BHW activity minutes ^3^	54	0.422 (0.268–0.575)	<0.001	0.619 *	<0.001
GIH-SED-Q	57	0.535 (0.360–0.711)	<0.001	0.602 *	<0.001

Kappa: weighted kappa coefficient; 95% CI: 95% confidence interval; Spearman’s rho: Spearman’s rho correlation coefficient. ^1^ Without ≥10 min bouts; ^2^ BHW-Q VPA × 2 + BHW-Q MPA (min/w); ^3^ BHW-Q VPA × 2 + BHW-Q MPA ^1^ (min/w); * *p* = 0.01.

**Table 5 ijerph-20-06579-t005:** Criterion validity and agreement for questionnaires of PA and SED using accelerometer.

Question	*n*	Kappa (95% CI)	*p* (k)	Spearman’s rho	*p* (rho)
Haskell-Q MPA	56	0.129 (−0.029–0.251)	0.029	0.407 *	0.002
Haskell-Q VPA	57	0.047 (−0.015–0.109)	0.146	0.139	0.301
Haskell-Q MPAtot ^1^	55	0.218 (0.038–0.398)	0.013	0.282 **	0.037
BHW-Q VPA	56	0.080 (−0.073–0.234)	0.243	0.256	0.057
BHW-Q MPA	56	0.341 (0.185–0.496)	<0.001	0.578 *	<0.001
BHW-Q MPAtot ^1^	55	0.237 (0.099–0.376)	<0.001	0.501 *	<0.001
Activity minutes ^2^	56	0.276 (0.141–0.411)	<0.001	0.429 *	<0.001
Activity minutes ^3^	54	0.254 (0.086–0.421)	0.002	0.385 *	0.004
GIH-SED-Q	57	0.116 (−0.022–0.254)	0.061	0.291 **	0.028

Kappa: weighted kappa coefficient; 95% CI: 95% confidence interval; Spearman’s rho: Spearman’s rho correlation coefficient. ^1^ Without ≥ 10 min bouts; ^2^ BHW-Q activity minutes vs. accelerometer activity minutes; ^3^ BHW-Q activity minutes ^1^ vs. accelerometer activity minutes ^1^ * *p* = 0.01 ** *p* = 0.05.

## Data Availability

Study data are not publicly available due to identifying patient data should not be shared. Upon reasonable request, de-identified data may be available from the corresponding author.

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
