# Peer review of "Test–Retest Reliability, Agreement and Criterion Validity of Three Questionnaires for the Assessment of Physical Activity and Sedentary Time in Patients with Myocardial Infarction"

_ijerph, 2023, doi:10.3390/ijerph20166579_

Round 1

Reviewer 1 Report

-          Introduction

Line 44-45: Moderate-intensity physical activity is 150-300 min/week and vigorous-intensity physical activity is 75-150 min/week?!

-          Methods

Line 93: Please provide some information about the number of participants, possible groupings, and etc.

Did you categorize the patients and perform subgroup analysis according to different criteria such as stenosis severity, type of treatment plan, or other features? How did you assure that all the enrolled participants were similar regarding MI characteristics?

-          Results

Line 190: Don't you think that this is a small sample size for conclusion?

Table 2: Where are the P values?

Table 2: In the last row under the column of “Test 2”, How is it possible that the minutes of activity with >10-min bouts was higher than without this criteria?

-          Conclusion

How did you conclude such “These PA questionnaires could be used in clinical practice as a screening tool and/or to evaluate the level of PA patients with an MI.” considering that the reliability, correlation, and agreement of the examined questionnaires were not high (lines 215-217).  

Reviewer 2 Report

The aim of this study is to assess the test-retest reliability, criterion validity and agreement of two physical activity (PA) and one SED questionnaire commonly used in clinical practice, developed by the Swedish National Board of Health and Welfare (BHW) and the Swedish national quality register SWEDEHEART. The authors conclude in this study that BHW-Q has a higher criterion validity than the Haskell-Q and that the test-retest reliability is similar for both questionnaires, and these PA questionnaires could be used in clinical practice as a screening tool and/or to evaluate the level of PA in patients with an myocardial infarction (MI). The reviewer thinks the results of this study to be useful clinical information for assessment of PA in patients with MI. However, there are several problems in this study.

Has the validity and reliability of GHW-Q, Haskell-Q, and GIH-SED-Q questionnaires been verified so far? Questionnaires are fully tested for validity and reliability as they are created. What's problem with these questionnaires? The reviewers cannot understand why the validity and reliability of these questionnaires should be re-verified.

In this study, the reviewer thinks the data have a small number of samples, and that the information is biased in assessing the adequacy of physical activity. It is necessary for the authors to objectively explain that the data in this study have a small number of samples and that the information is biased.

In this study, the authors set the MI patients to wear the accelerometer for 7 days. The reviewer considers the duration of accelerometer wear to be extremely short to assess the physical activity in MI patients. In this study, physical activity can be evaluated for only 5 days, excluding the wearing and removing days. Data for at least 7 days, including weekdays, weekends, and holidays, excluding the wearing and removing days, are required to evaluate the physical activity. The reviewer believes that accelerometer measurements to be extremely important to correctly assess the validity of the physical activity questionnaire.

In this study, information on the physical activity in MI patients is poor. Information on the patient's physical activity level is extremely important in this study. The authors should add the information of accelerometer-assessed patient’s physical activity in Table 2.

In the 'Results' section (P7, L206-207 and L209-211), the authors insufficiently explain Tables 3 and 4. The authors should carefully explain these results.

Reviewer 3 Report

This paper evaluated the test-retest reliability, criterion validity and agreement of the BHW-Q, Haskell-Q and GIH-SED-Q in patients after an MI. The authors concluded that the BHW-Q has a higher criterion validity than the Haskell-Q and that the test-retest reliability is similar for both questionnaires. The manuscript was well-written. Studies about the management of PA in patients with MI is needed to guide current clinical care. I only have some minor suggestions for the authors.

1. The sample size of the study seems small.  Please discuss more about it.

2. “The proportion of women included in this study reflects the general 290 MI population in Sweden.” Please add relevant references here.

Round 2

Reviewer 2 Report

The reviewer thinks that the revised manuscript generally responds well to the reviewers' comments. However, the following points still need to be revised.

The authors explain that the number of patients is sufficient, but please provide the basis for this reason (sample size etc.).

In the 'Results' section, the authors insufficiently explain Tables 3 and 4. Reviewers can understand not to duplicate the results, but the current manuscript is insufficient. The reviewer thinks that an explanation is necessary, even if it is concise.

Author Response

We thank the reviewer for the additional comments. We have now made the requested changes accordingly.

The authors explain that the number of patients is sufficient, but please provide the basis for this reason (sample size etc.).

Answer: Thank for this comment. In this kind of methodological studies, it is not possible to perform sample size calculations. The sample size in our study is comparable to many other similar studies. It is recommended to have between 50-100 subjects, yet the most important factor is having subjects in all parts of the scale, i.e low to high physically active. To be transparent, we have changed the following sentence in the discussion: “The sample size is comparable to validation studies and included a variety of physical activity levels. Yet, not so many subjects with very low physical activity levels. A replication of the results using larger sample and including subjects with a broader range of activity level could be beneficial”

In the 'Results' section, the authors insufficiently explain Tables 3 and 4. Reviewers can understand not to duplicate the results, but the current manuscript is insufficient. The reviewer thinks that an explanation is necessary, even if it is concise.

Answer: Thanks for the comment. We have added an explanation to the tables. Table 3 is described in the text above the table. The following text has been added in the results:

“Test-retest reliability was moderate for Haskell-Q MPA (k = 0.434), BHW-Q VPA (k = 0.521), “Activity minutes” without ≥ 10 minutes bouts (k = 0.442) and GIH-SED-Q (k = 0.535). The remaining questions presented a fair test-retest reliability (k = 0.219-0.375) (Table 4). “

“Correlation between PA and SED questions and categorical accelerometer data was fair for Haskell-Q MPA (r = 0.407), BHW-Q MPA (r = 0.578), BHW-Q MPAtot (r = 0.501) and “Activity minutes” (r = 0.429). There, were a fair agreement for BHW-Q MPA and BHW-Q MPAtot (k = 0.341 resp. 0.237). All other questions presented a slight to poor correlation to categorical accelerometer data (r = 0.139-0.291) (Table 5).”